

# Tandem configuration of differential mobility and centrifugal particle mass analyzers for investigating aerosol hygroscopic properties

Sergey S. Vlasenko[1], Hang Su[2], Ulrich Pöschl[2], Meinrat O. Andreae[2], and Eugene F. Mikhailov[1,2]

[1]Department of atmospheric physics, Saint-Petersburg University, St. Petersburg State University, SPbSU, SPbU, 7/9 Universitetskaya nab., St. Petersburg, 199034, Russia
[2]Biogeochemistry and Multiphase Chemistry Departments, Max Planck Institute for Chemistry, Hahn-Meitner-Weg 1, 55128, Mainz, Germany

*Correspondence to*: Eugene F. Mikhailov (eugene.mikhailov@spbu.ru)

**Abstract.** A tandem arrangement of Differential Mobility Analyzer and Humidified Centrifugal Particle Mass Analyzer (DMA-HCPMA) was developed to measure the deliquescence and efflorescence thresholds and the water uptake of submicron particles over the relative humidity (RH) range from 10% to 95%. The hygroscopic growth curves obtained for Ammonium sulfate and sodium chloride test aerosols are consistent with thermodynamic model predictions and literature data. The DMA-HCPMA system was applied to measure the hygroscopic properties of urban aerosol particles, and the kappa mass interaction model (KIM) was used to characterize and parameterize the concentration-dependent water uptake observed in the 50–95% RH range. For DMA-selected 160 nm dry particles (mass of 3.5 fg), we obtained a volume-based hygroscopicity parameter, $\kappa_v \approx 0.2$, which is consistent with literature data for freshly emitted urban aerosols.

Overall, our results show that the DMA-HCPMA system can be used to measure size-resolved mass growth factors of atmospheric aerosol particles upon hydration and dehydration up to 95% RH. The direct measurements of humidified
particle mass allow avoiding complications that occur in the commonly used mobility-diameter-based HTDMA technique due to poorly defined particle morphology and density.

**Keywords:** atmospheric aerosol, hygroscopic growth, centrifugal particle mass analyser.

## 1 Introduction

The interaction of atmospheric aerosol particles with water vapor results in size changes which strongly affect the optical
properties of the aerosol particles and consequently their direct radiative effect on the Earth's energy budget (Waggoner et al., 1981; Rastak et al., 2014, and references therein). Furthermore, hydrophilic aerosol particles are able to act as cloud condensation nuclei, thereby modifying the radiative properties and lifetime of clouds (indirect aerosol effects) (Twomey, 1977; Haywood and Boucher, 2000). Additionally, hygroscopicity is an important factor in the chemical reactivity of aerosols (Shiraiwa et al., 2013). Therefore, the hygroscopic properties of aerosol particles have been of interest throughout
the whole history of aerosol study, with respect to both the direct description of their hygroscopic growth and the influence



on their optical parameters ( Orr et al.,1958; Hanel, 1976; Rader and McMurry, 1986; Berg et al., 1998; Cheng et al. 2008; Fierz-Schmidhauser et al., 2010ab; Zieger et al., 2013).

Humidified Tandem Differential Mobility Analyzers (HTDMA) are the most commonly used technique to study the hygroscopic behavior of size-selected aerosol particles (McMurry and Stolzenburg, 1989; Brechtel and Kreidenweis 2000a; Gysel et al., 2002; Mikhailov et al., 2004; Eichler et al., 2008; Swietlicki et al., 2008). The first DMA selects particles of a specific size from the previously dried polydisperse aerosol. Then this monodisperse aerosol is humidified to a set RH. The number size distribution of the humidified sample is then measured by a second DMA operated at the same RH as the humidified sample. The HTDMA method is used for both laboratory and field measurements, and typically covers the particle mobility diameter in the range of 6 - 300 nm. However, this instrument classifies particles according to their electrical mobility, and therefore the water uptake is calculated indirectly with an uncertainty introduced by lack of knowledge about particle shape and density (DeCarlo et al., 2004; Gysel et al, 2004; Mikhailov et al., 2004).

A number of alternative detectors have been employed to replace the second DMA to probe the hygroscopic properties of the size-selected atmospheric aerosol. Several studies used an optical particle counter (OPC) (Covert et al., 1990; Hering and McMurry, 1991; Brand et al., 1992; Kreisberg et al., 2001). Sorooshian et al. (2008) proposed a differential aerosol sizing and hygroscopicity spectrometer probe (DASH-SP), which employed four OPCs to measure the hygroscopic growth at three RHs simultaneously and thereby reduces measurement time. Massling et al. (2007) applied a hygroscopicity differential mobility analyzer-aerodynamic particle sizer system (H-DMA-APS) to measure the hygroscopic properties of aerosol particles at or near 1μm in dry diameter during ACE-Asia. Some other methods deal with polydisperse aerosol particles. Snider and Peters (2008) and Hegg et al. (2007) obtained growth factors by comparing OPC size distributions measured at different RHs. Many researchers employed humidity controlled nephelometers and estimated growth factors from variations of aerosol light scattering coefficients with changing RH (Magi and Hobbs 2003; Kim et al. 2006; Fierz-Schmidhauser et al. 2010a,b). All of these methods rely on certain assumptions (particle morphology, density, refractive index, etc.) to convert optical or mobility growth factor measurements into a mass-based scale.

However, mass-based measuring techniques are more appropriate to describe the water uptake of aerosols in conjunction with thermodynamic or parametric models (Mikhailov et al., 2013). The single particle levitation technique is commonly used for the direct measurement of the water vapor uptake/release by particles due to RH variations (Tang and Munkelwitz, 1993, 1994; Peng and Chan, 2001). This technique enables high precision measurements of the mass growth factor in hydration and dehydration mode, but it is applicable only to supermicron (typically 5-25 μm) particles and can be used only in laboratory conditions. The other mass-based techniques employ aerosol sampled on filters (Lee and Hsu, 1998; Mikhailov et al., 2011). This approach is suitable for measurements of ambient aerosol, but only in off-line mode, and therefore, the loaded filters require careful handling to avoid possible particle mass loss between sampling and measuring.

The most reliable online technique for measuring submicron particle mass is based on the Ehara aerosol particle mass analyzer (APM; Ehara et al. 1996) in combination with DMA. The early studies based on this technique were used to determine the density and shape factor of size-selected test particles and atmospheric aerosols (McMurry et al. 2002; Geller





et al. 2006). In the more recent publications, the DMA+APM technique was used to follow the aging of soot particles and the change of effective density due to condensation of organic (oleic acid and anthracene) (Slowic et al., 2007) and inorganic (sulfuric acid and water) species (Pagels et al. 2009; Johnson et al., 2015).

We present here an application of the DMA-HCPMA technique to study the hygroscopic growth of aerosol particles. A

hygroscopicity centrifugal particle mass analyzer (HCPMA) was employed for direct measurements of particle mass increase/decrease due to water uptake/release. The concept has been briefly introduced in Vlasenko and Mikhailov (2013). In this paper, we present a detailed description of the experimental procedure and results obtained for both laboratory-generated particles and ambient atmospheric aerosols. Our work is mainly focused on the applicability of the tandem DMA-HCPMA setup for investigating the hydration/dehydration of aerosol particles, including deliquescence/efflorescence phase

transitions.

## 2 Experimental

A sketch of the HCPMA system is shown in Figure 1. Aerosols were generated by nebulization of aqueous solutions of the investigated pure substances in deionized water (18.2 MΩ·cm, Millipore – Milli-Q plus 185) with a solute mass fraction

0.01%, using a constant output atomizer (TSI, model 3075). The following reference substances were used to prepare test aerosols for the proposed method: ammonium sulfate ($(NH_4)_2SO_4$, Fluka, >99.5%) and sodium chloride (NaCl, Merck, > 99.5%,). The solution droplets were dried using a silica gel diffusion dryer (SDD) with aerosol residence time ~ 10 s; the residual relative humidity was <10% throughout all experiments. The dry polydisperse aerosol was passed through a neutralizer (NL) (X-ray, Model TSI 3087), and a near-monodisperse aerosol with the desired initial dry particle mobility

diameter ($D_b$) was selected by the differential mobility analyzer (DMA, Model TSI 3081). The resulting monodisperse particles were then flowed at 0.5 l min$^{-1}$ through a series of one or two single-tube Nafion humidity exchangers.

In hydration mode, the outer tube of the humidity exchanger (2.4 m long) was supplied with humidified air at controlled RH. The RH was increased stepwise from 15 to 95% by mixing of dry air with an airflow saturated with water vapor, varying the ratio of humidified to dry air to produce a total flow of 2 l min$^{-1}$. Water-saturated air was produced by bubbling clean and dry

filtered air (TSI 3074B) through water heated to 36 °C (Mikhailov et al., 2004). The hydration mode provides information about deliquescence phase transitions of dry particles and the hygroscopic growth of deliquesced particles (aqueous solution droplets) as a function of relative humidity. The aerosol residence time in the conditioner and subsequent lines leading to the CPMA is ~10 s.

In dehydration mode, a series of two Nafion humidity exchangers was used for aerosol RH conditioning. The outer tube of

the first one (1.2 m long) was filled with water, which caused the particles to deliquesce and form aqueous droplets. The outer tube of the second one (2.4 m long) was supplied with air at controlled RH that decreased stepwise from 95 to 15% RH. The dehydration mode allows studying the hysteresis effect and efflorescence of aerosol particles.





The relative humidity (RH) and the temperature (T) of the aerosol flow at the inlet and outlet of the CPMA were measured with capacitive humidity sensors (accuracy ±2% RH) and temperature sensors (accuracy ± 0.1 K) (ALMEMO 2390, Ahlborn FH A646-E1C).

The heart of the apparatus is a centrifugal particle mass analyzer (CPMA, Cambustion Ltd) designed for classifying aerosol

particles according to their masses. The analyzer consists of two rapidly spinning coaxial cylindrical electrodes. The initially charged aerosol particles pass through the annular inter-electrode space, where they experience centrifugal and electrostatic forces acting in opposite directions (Ehara et al. 1996). Depending on the rotation speed and voltage applied, these forces are balanced for aerosol particles with certain mass-to-charge ratio. These particles move through the analyzer without precipitating on the electrodes. The other particles are forced either to the inner or to the outer electrode and adhere to their

surfaces. Thus, a CPMA selects particles with a specific mass, provided that the charge on the particles is the same and known. To improve the transfer function of the classifier, the inner electrode rotates slightly faster than the outer one, producing a stable system of forces (Olfert and Collings, 2005). The particle mass analyzer was operated in the step-by-step scanning mode, where rotation speed and applied voltage are varied in a discrete way to scan the desirable particle mass range. The CMPA, in conjunction with the condensation particle counter (CPC) (TSI model 3787), measured the particle

number mass distributions as a function of the applied RH history.

The main experimental challenge was caused by heat generation inside the CPMA due to the friction generated by the rotating cylinders and the heat produced by the electric motor. This led to a gradual warming of air passing through the CPMA and a corresponding RH decrease. To minimize the difference in RH in the CPMA analyzer, the Nafion humidity exchangers and connecting tubes were immersed into a water bath circulator with controlled temperature to provide a

gradual heating of the humidified aerosol flow in agreement with the CPMA internal temperature growth. Additionally, a humidified aerosol flow (0.5 l min$^{-1}$) was mixed in downstream of the Nafion Conditioner Air (NCA) flow (1 lmin$^{-1}$) (Fig. 1). Using these operation modes, it was possible to ensure an agreement between the RHs at the inlet and outlet of the CPMA within 2% RH.

The dry aerosol particle mass was determined at the beginning of every measurement cycle. For this purpose, the DMA-

selected particles were directly entered into the CPMA, bypassing the humidity exchangers. Then, the aerosol flow was redirected through the humidifier system with preset RH, and, after stabilization of the aerosol RH, a spectrum of particle concentration versus mass of the humidified aerosol was measured by the CPMA. The total CPMA scanning time over a measurement cycle depends on the desired mass range and resolution. Due to mass distribution broadening with increasing RH, more scanning steps are needed to provide the same CPMA mass spectra resolution over the entire RH range. In this

study, the CPMA scanning time varied within 10-20 min. The lower value refers to the dry aerosol conditions.

The hygroscopic mass growth factor was calculated as the relative particle mass increase due to water uptake:

$$G_m = \frac{m_w + m_d}{m_d},$$    (1)





where $m_d$ is the mass of the dry particles and $m_w$ is the mass of water in the wet particles. Figure 2 is an example of the ammonium sulfate number distribution versus particle mass measured at 10%, 77%, and 85% RH, respectively. It shows that upon hydration the initial narrow mass spectral density of the dry aerosol particles ($m_d$=0.18 fg) became broader, and the maximum of the distribution shifted to a larger mass range. The observed plateau with a minor peak on the initial

distribution (blue curve) arises from double-charged particles (+2e) (Symonds et al., 2011).

The modal mass of the respective particle mass distribution was used for calculation of the $G_m$ value, as specified in Fig. 2. The precision of the particle mass measurements was ± 5% (relative standard deviation of repeated measurements), which translates into a ~7% uncertainty in $G_m$.

**3 Results and discussion**

**3.1 Ammonium sulfate and sodium chloride mass growth**

Ammonium sulfate (AS) and sodium chloride (NaCl) were chosen as reference substances to calibrate the HCPMA setup since their hygroscopic properties are well known from measurement (Gysel et al., 2002; Mikhailov et al. 2004, 2009;

Kreidenweis et al, 2005; Biskos et al., 2006a, b; Sorooshian et al., 2008) and theory (Clegg et al., 1998 a,b; Martin, 2000; Topping et al., 2005; Mikhailov et al., 2013; Cheng et al. 2015). Figure 3 shows the results obtained for AS and NaCl particles with initial mobility diameter $D_b$=60 nm and dry masses of 0.18±0.01 fg and 0.21±0.01 fg, respectively. It illustrates the typical behavior of crystalline inorganic salt aerosol particles interacting with water vapor (Martin, 2000).

Upon hydration, the deliquescence transition results in a stepwise increase of the HCPMA-derived mass at RH=80±2% for

ammonium sulfate and at RH=75±2% for sodium chloride, respectively. The obtained deliquescence relative humidity (DRH) is in good agreement with literature data of DRH for crystalline AS: 79.9±0.5% and 75.3±0.1% for NaCl (Gysel et al., 2002; Seinfeld and Pandis, 2006; Mikhailov et al., 2009, and references therein). The measured mass growth factors are in agreement with the "full Köhler model" (Brechtel and Kreidenweis, 2000b; Rose et al., 2008; Mikhailov et al., 2009) based on the water activity parameterization derived from the Aerosol Inorganics Model (AIM, Clegg et al, 1998a) .

Averaged over the whole range of 40–95% RH, the mean relative deviations between measurement and model results were within 10%. It should be noted that a 10% deviation in mass growth factor corresponds to 3% accuracy in size growth factor, which is typical for HTDMA-derived data (Duplissy et al, 2009). For NaCl particles at RH above the DRH, the measured $G_m$ values are systematically below (by ~ 5%) than predicted by theory. Most likely the observed discrepancy was caused by the short residence time in the humidifier system (~10 s), so that the sodium chloride droplets had not yet reached their

equilibrium masses (Cruz and Pandis, 2000; Chan and Chan, 2005; Duplissy et al. 2009). For both AS and NaCl particles, intermediate growth factors between dry and deliquesced particles were observed (Fig.3 – blue crosses). This looks like the



apparently non-prompt phase transition that was previously described by Mikhailov et al. (2004) and Biskos et al. (2006a). The most reasonable explanation for the observed effect is due to RH variability inside the CPMA. The 2% RH uncertainty is too high to accurately resolve the particles' deliquescence point. The RH appears to vary slightly while the aerosol particles pass through the CPMA. Thus, when measuring near the deliquescence point, the initially solid particles may

deliquesce somewhere inside CPMA if the local RH exceeds the DRH. These inner deliquescence events lead to observation of transitional particle mass spectra (Fig.4a), which seems to be an artifact.

In dehydration mode, both ammonium sulfate and sodium chloride particles reveal hygroscopic hysteresis, but efflorescence of salt solution droplets in the CPMA experiments occurred as a rule at much higher RH than the reference efflorescence relative humidity (ERH): 35±2% for AS and 43±3% for NaCl (Seinfeld and Pandis, 2006). Approximately in half of the

measurements, an abrupt decrease of solution droplet mass was observed at RH=65±5% for AS and at RH=60±3% for NaCl, respectively. In this case, the peaks of effloresced and non-effloresced particles were merged into a unimodal distribution with intermediate modal mass as illustrated in Fig. 4b. These mass distributions account for the transitional growth factor values observed during efflorescence (Fig. 3, red crosses). Although both deliquesced and effloresced transitional spectra are unimodal (Fig. 4a and Fig. 4b, respectively), their mass distribution shape and width are different. The hydration transitional

spectra are narrower than the dehydration ones (Fig.4c), which is due to the irreversibility of the particles' efflorescence. As a result, the output aerosol in dehydration mode is a mixture of droplets and dry particles, while in hydration mode it consists mainly of droplets. It is important to note that any change of the particle mass inside the CPMA leads to a deflection of the particle trajectory due to the disturbance of the force balance. Therefore, all the transient mass spectra should be considered unreliable.

The spatial variability of RH inside the CPMA induced by the above-mentioned frictional warming appears to be the most likely cause of the observed efflorescence at RH close to the DRH. It is reasonable to suppose that the relative humidity is minimal close to the electrode surface, and that at RH<DRH salt crystals may form first on the electrode surface. The gap between the two cylinders is only ~1 mm, and due to force imbalance, some metastable droplets could make contact with crystals on the surface, triggering an efflorescence transition as specified in Fig. 5. Since the mass of these newly formed

effloresced particles is smaller than that of the initial droplets, the electrostatic forces ($F_e$) will prevail over the centrifugal forces ($F_c$), and the dry particles either will move towards the inner electrode or pass through the gap as shown in Fig. 5. In the latter case, the CPC will count these particles as droplets with the preset mass, thus distorting the actual mass distribution (Fig. 4b).

Contact efflorescence is especially effective if the seed crystal is of the same composition as the anticipated crystalline

phase. In this case, due to the isochemical collision, no activation barrier is required for nucleation and efflorescence will occur at ERH close DRH. Contact efflorescence experiments by Davis et al. (2015) have shown that upon collision of $AS_{solid}$ with $AS_{droplet}$ and $NaCl_{solid}$ with $NaCl_{droplet}$, the efflorescence of the metastable droplets occurred at 79±2% RH for AS and at 74±2% RH for NaCl, respectively. In contrast, heterochemical collision did not significantly influence ERH. For example,





single contact of a metastable AS droplet with dry NaCl particles induced AS crystallization at ERH =38±2%, that is only by ~3% RH higher than homogeneous ERH (~35%).

In some cases, the particle mass distribution measured by CPMA bifurcated into a bimodal one, indicating a partial efflorescence of solution droplets. Figure 4d shows the particle number mass distribution measured at relative humidities

close to the efflorescence threshold upon dehydration of the AS particles. The first distribution mode corresponds to the mass of the initial dry particles ($m_d$ =0.18 fg) and represents effloresced particles. The second mode represents non-effloresced droplets. Its position is shifted to larger particle masses and depends on RH. The ratio of the peak heights depends on RH as well, i.e., the lower the RH, the more the effloresced particle mode becomes predominant as compared to the non-effloresced one.

Since in this case effloresced and non-effloresced peaks are resolved (Fig. 4d), their measured masses were used to calculate the mass growth factor of the salt particles (Eq. (1)). This makes it possible to follow the dehydration mode down to RH=42±2% for sodium chloride (Fig. 3a) and RH=37±2% for ammonium sulfate (Fig. 3b) particles, respectively. For NaCl particles the observed efflorescence relative humidity is in good agreement with the HTDMA-derived values of ERH =44±2.5% obtained by Biskos et al. (2006b) for 60 nm particles. For AS, the HCMPA-derived value is at the upper end of

the literature ERH values ranging from 30% to 37% RH, reported for particles ≤100 nm (Biskos et al., 2006a; Badger et al., 2006; Sjorgen et al.,2007; Mikhailov et al., 2009).

### 3.2 Ambient aerosol measurements

The DMA-HCPMA setup was used to determine the hygroscopic properties of ambient atmospheric aerosol particles. The

sampling site (59°88′ N, 29°82′ E) was located in a suburban forest environment at Petrodvoretz, about 35 km southwest of Saint Petersburg, Russia (SPB sample). The measurements were carried out in the daytime on 25 and 26 March 2014. The ambient aerosols were first dried by a silica gel diffusion dryer to a residual RH <10% and then entered to the DMA-HCMPA system. The measurement procedure was the same as used for single component solutes. During the sampling campaign, the particle mass distribution was relatively stable. A modal mobility diameter of $D_b$ = 160 nm was selected for

the hygroscopic growth measurements, which corresponds to $m_d$=3.5 fg. The precision of the particle diameter measurements was generally about ±10% and the total uncertainty of the mass growth factor was estimated to be 14%. Concurrent chemical analysis of the aerosol fractions in the size range of 20-300 nm indicated that the SPB sample consisted mostly of inorganic ions, generally in the form of ammonium sulfate ($f_{AS}$~0.45), and organics ($f_{OC}$~0.52) (Mikhailov et al., 2016). Inserting these mass fraction values ($f_i$) and densities of $\rho_{OC}$=1.4 g cm$^{-3}$ (Kostenidou et al., 2007), $\rho_{AS}$=1.77 g cm$^{-3}$

(Lide, 2005) in Eq. (2) yields an average weighted bulk density of $\rho_d$ =1.6 g cm$^{-3}$





$$\rho_d = \left( \sum_i \frac{f_i}{\rho_i} \right)^{-1} \tag{2}$$

Figure 6a shows the mass growth factors, determined as a function of relative humidity upon hydration and dehydration. The onset of deliquescence was noticeable at ~63% RH and then water uptake gradually increased, reaching a mass growth factor value of 2.2 at ~93% RH. Upon dehydration, the mass growth factors were slightly larger than those observed for the hydration mode, and at ~62% RH particles underwent efflorescence. Both deliquescence and efflorescence transitions were not pronounced. Most likely, the observed reversible stepwise particle mass change in the hydration and in the dehydration experiments was related to a phase transition between collapsed and swollen semi-solid organic structures (Mikhailov et al., 2009). In the framework of the κ-mass interaction model (KIM), this transition can be regarded as a quasi-eutonic threshold between metastable amorphous phases (Mikhailov et al., 2013).

Figure 6b shows the mass-based hygroscopicity parameter ($\kappa_m$) plotted against the mass growth factor. The concentration dependence of $\kappa_m$ was calculated as specified in Mikhailov et al. (2013). Briefly, in analogy with the volume-based hygroscopicity parameter (Petters and Kreidenweis, 2007), we define a mass-based hygroscopicity parameter, $\kappa_m$:

$$\frac{1}{a_w} = 1 + \kappa_m \frac{m_d}{m_w} \; , \tag{3}$$

where $a_w$ is the activity of water, $m_d$ is the total mass of the dry particle material, and $m_w$ is the mass of water in the wet particle (aqueous droplet). By combining Eq. (1) and Eq. (3) we obtain

$$a_w = \left( \frac{\kappa_m}{G_m - 1} + 1 \right)^{-1} . \tag{4}$$

Based on Eq. (4) an approximate mass-based $\kappa_m$–Köhler equation can be written as follows (Mikhailov et al., 2013):

$$\frac{RH}{100\%} \approx \left( \frac{\kappa_m}{G_m - 1} + 1 \right)^{-1} \exp \left( \frac{4\sigma_w M_w}{RT\rho_w} \left[ \frac{\pi\rho_w}{6 G_m m_d} \right]^{1/3} \right) \tag{5}$$

where $M_w$, $\sigma_w$, and $\rho_w$ are the molar mass, surface tension, and density of pure water, $R$ is the universal gas constant, $T$ is the temperature. From the measurement of the $G_m(RH)$ data we derived the $\kappa_m$ values (Fig. 6b) using Eq. (4) and Eq. (5) at $m_d$=3.5 fg.

For mixed organic-inorganic particles, KIM describes three distinctly different regimes of hygroscopicity: (I) a quasi-eutonic deliquescence and efflorescence regime at low-humidity, where substances are just partly dissolved and exist also in a non-dissolved phase, (II) a gradual deliquescence and efflorescence regime at intermediate humidity, where different solutes undergo gradual dissolution or solidification in the aqueous phase; and (III) a dilute regime at high humidity, where the



solutes are fully dissolved approaching their dilute hygroscopicity. In each of these regimes, the concentration dependence of $\kappa_m$ can be described by simplified model equations:

Regime I:

$$\kappa_m = k_1 \left(G_m - 1\right) \tag{6}$$

Regime II:

$$\kappa_m = k_2 + k_3\left(G_m - 1\right) + k_4\left(G_m - 1\right)^{-1} + k_5\left(G_m - 1\right)^{-2} \tag{7}$$

Regime III:

$$\kappa_m = k_5\left(G_m - 1\right)^{-2} + k_6 \quad . \tag{8}$$

Here $k_1$ to $k_6$ are fit parameters related to the solubility and interaction coefficients of all involved chemical components

(Mikhailov et al., 2013; Eqs. (39-44)). In the dilution mode (III), $\kappa_m$ decreases with increasing $G_m$ and becomes concentration independent at very high values of $G_m$ (Eq.(4)). According to Eq. (8), the fit parameter $k_6$ can be regarded as the dilute hygroscopicity parameter of the investigated sample of particulate matter ($\kappa_m^0$). In this experiment, the dilution mode (III) was not clearly pronounced, generally because in the DMA-HCPMA setup the upper RH value does not exceed ~95%. Therefore, only quasi-eutonic deliquescence (Ia)/efflorescence (Ib) and gradual deliquescence (IIa)/efflorescence (IIb)

regimes have been considered by using KIM model Eq. (6) and Eq. (7) to fit the data points (Fig. 6b, solid lines). The obtained best-fit parameters $k_1$- $k_5$ are listed in Table 1.

By inserting KIM-derived values of $\kappa_m$ and $G_m$ in Eq. (5), we obtained the $G_m$(RH) dependences displayed in Fig. 6a (solid lines). It can be seen that the model curves are in good agreement with the measurement data. Note that, for the quasi-eutonic regime (I), the combination of Eq. (4) and Eq. (6) yields a constant water activity value given by $a_w = (k_1 + 1)^{-1}$. This

relation yields the following quasi-eutonic RH values characterizing the deliquescence (Ia) and efflorescence (Ib) phase transitions: 62.7% and 61.9%, respectively (Fig. 6a,b).

Under the volume-additivity assumption, the mass-based parameter, $\kappa_m$, can be converted to the Petters and Kreidenweis (2007) volume-based parameter, $\kappa_v$, by the relation:

$$\kappa_v = \kappa_m \frac{\rho_d}{\rho_w} \tag{8}$$

We chose for the calculation of $\kappa_v$ the $\kappa_m$ fit value of 0.11±0.04, which is the intercept of the model curves at $G_m$ =2.1 (Fig. 6b). This $\kappa_m$ value corresponds to the most dilute solution

concentration in the particles achieved in the given experiment. Inserting $\kappa_m$ = 0.11±0.04 and the estimated density $\rho_d$ =1.6 g cm$^{-3}$ in Eq. (8) yields $\kappa_v$ = 0.18±0.08. The obtained value is in reasonable agreement with the CCNC-derived $\kappa_v$ = 0.22±0.12 averaged from 21 to 31 March 2014 measured at the same site for 100 nm particles (Mikhailov et al., 2016) and $\kappa_v$ ~ 0.2

reported by Zhang et al. (2014) and Rose at al. (2010) for freshly emitted aerosols from urban pollutants. Most likely, the



low $\kappa_v$ value we obtained arises from mixed particles with coatings by hydrophobic organics, produced from fossil fuel combustion and biomass burning (Andreae and Rosenfeld, 2008).

## 4 Conclusion

A new DMA-HCPMA technique for measuring the hygroscopic properties of laboratory and ambient aerosols is introduced. Laboratory tests with inorganic compounds were conducted to verify the proposed technique in hydration and dehydration modes. Ammonium sulfate and sodium chloride particles were used as reference inorganic aerosols. A fairly good agreement was observed between measured mass growth factors and those calculated with a full Köhler model. The difference between experimental results and theoretically predicted liquid particle growth factor values does not exceed 10%. The measured DRHs of ammonium sulfate and sodium chloride aerosols are in agreement with literature values within 2% RH uncertainty. In the dehydration experiment, efflorescence occurred at higher RH than the ERH of homogenous nucleation. This effect appeared to be associated with contact efflorescence initiated by collision between metastable micro-droplets and salt crystals deposited on the electrode surface. We suggest that under controlled composition of particles on the electrode surface, the HCPMA system could be additionally used to study isochemical and heterochemical contact efflorescence.

The DMA-HCPMA tandem system was also applied to measure mass growth factors of urban aerosol particles. The kappa mass interaction model (KIM) was used to characterize and parameterize non-ideal solution behavior and concentration-dependent water uptake by atmospheric aerosol samples in the 50–95% RH range. Overall, both test results and field measurements have shown that the DMA-HCPMA system described above can be applied for aerosol size-resolved mass growth factor measurements in hydration and dehydration modes up to 95% RH.

*Acknowledgements.* This work was supported by the Max Planck Society (MPG), RFBR grants 16-05-00718 and 16-05-00717, and Saint Petersburg State University (SPBU) grant BRICS 11.37.220.2016. We thank the Geomodel Research Center at Saint Petersburg State University for help with chemical analysis of the ambient aerosol samples.

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



**Table 1.** KIM fit parameters for the SPB sample with dry particle mass of 3.5 fg ($D_b$ = 160 nm). The columns $n$ and $R^2$ give the number of data points and the coefficient of determination of the fit.

| Regime | $n$ | $R^2$ | Fit equation | Best fit parameter ± standard error |
|---|---|---|---|---|
| Quasi-eutonic deliquescence (Ia) | 11 | 0.89 | 6 | $k_1 = 0.61 \pm 0.04$ |
| Gradual deliquescence (IIa) | 60 | 0.63 | 7 | $k_2 = 0.33 \pm 0.01$ <br> $k_3 = -0.007 \pm 0.001$ <br> $k_4 = -0.009 \pm 0.0002$ <br> $k_5 = 2.35 \times 10^{-4} \pm 1.08 \times 10^{-4}$ |
| Gradual efflorescence (IIb) | 62 | 0.66 | 6 | $k_2 = 0.18 \pm 0.02$ <br> $k_3 = -0.049 \pm 0.017$ <br> $k_4 = -0.017 \pm 0.005$ <br> $k_5 = 5.67 \times 10^{-4} \pm 3.42 \times 10^{-4}$ |
| Quasi-eutonic efflorescence (Ib) | 11 | 0.91 | 7 | $k_1 = 0.63 \pm 0.03$ |





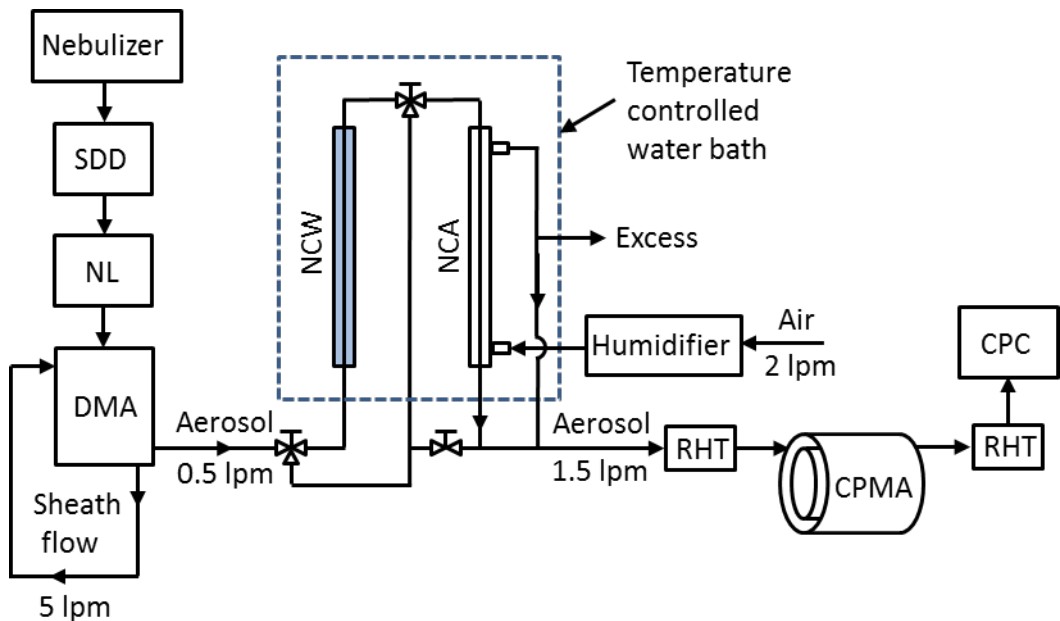

**Figure 1.** Experimental setup of the hygroscopicity centrifugal particle mass analyzer (HCPMA) system: SDD – silica gel diffusion dryer, NL – aerosol neutralizer, DMA – differential mobility analyzer, NCW – Nafion conditioner with water, NCA- Nafion conditioner with air, RHT- relative humidity and temperature sensor, CPMA - centrifugal particle mass analyzer, CPC- condensation particle counter.





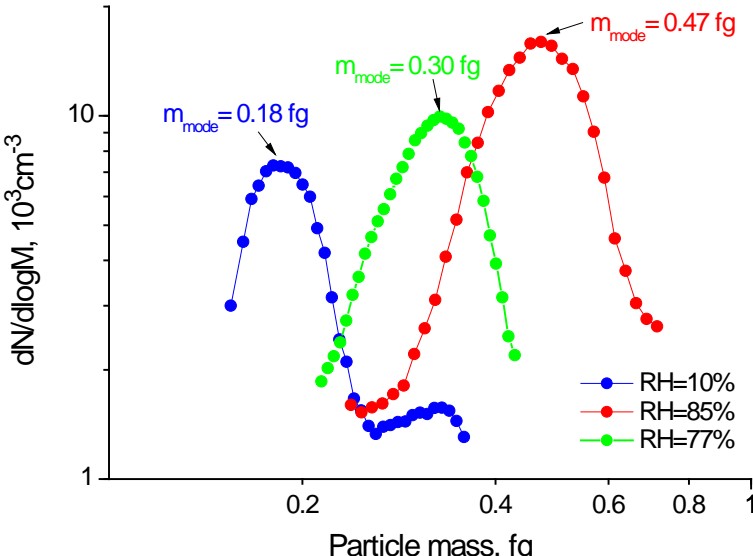

**Figure 2** HCPMA measured particle number mass distribution of ammonium sulfate at different RH with initial dry particle mass $m_d$ = 0.18 fg. The indicated $m_{mode}$ is the modal value of the particle mass distribution used for the mass growth factor ($G_m$) calculation.





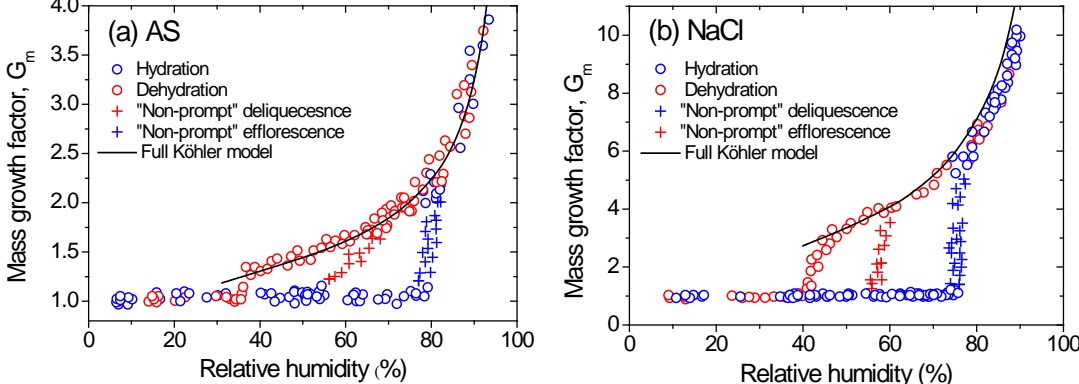

**Figure 3.** The mass growth factor ($G_m$) of ammonium sulfate **(a)** and sodium chloride **(b)** aerosol particles observed as a function of relative humidity (RH) compared to the full Köhler model: blue and red crosses represent apparent non-prompt deliquescence and efflorescence thresholds, respectively.





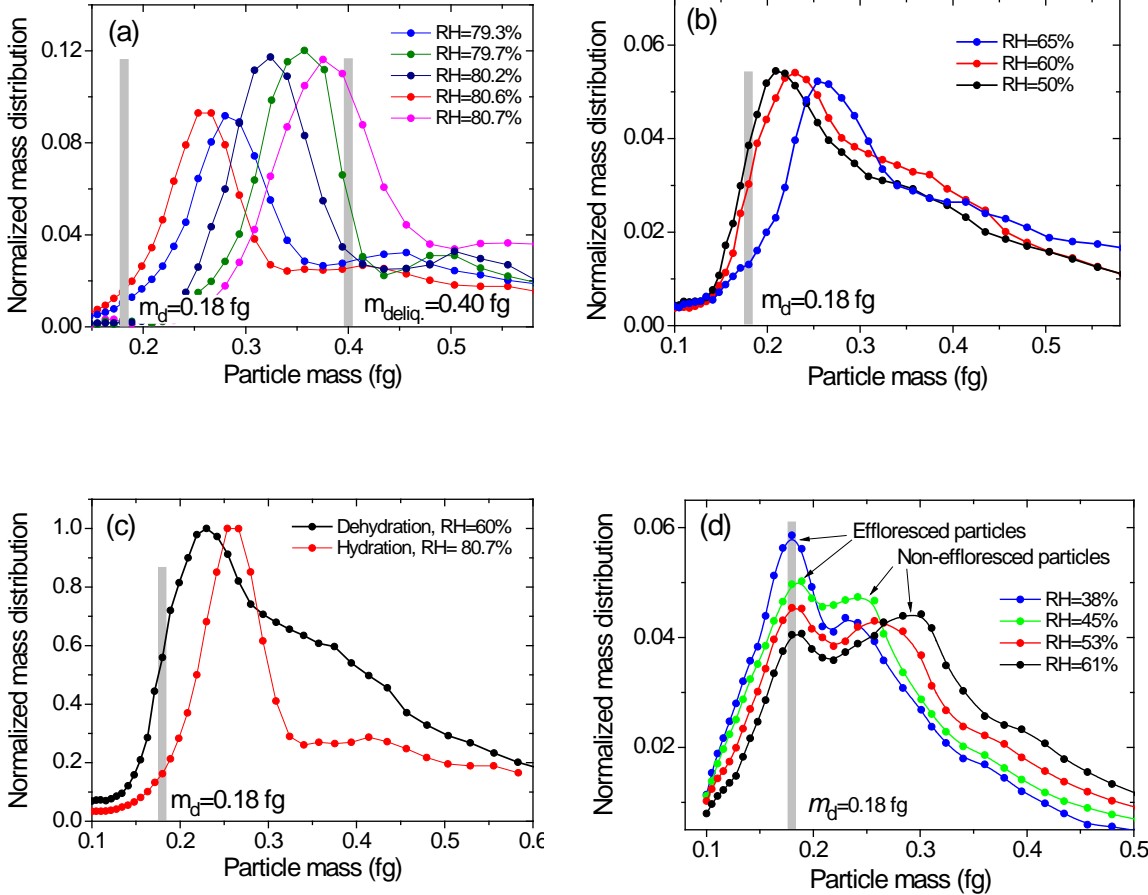

**Figure 4.** Normalized particle number mass distributions measured at relative humidities close to the deliquescence and
efflorescence thresholds of AS with $m_d$ = 0.18 fg: **(a)** unimodal intermediate mass distributions observed near the DRH,
$m_{deliq.}$=0.40 fg is the mass of fully deliquesced particles; **(b)** unimodal and **(d)** bimodal mass distributions observed upon
particle dehydration at RH< 65%; **(c)** mass distribution broadening upon particle hydration (red), and dehydration (black).
The RH uncertainty was estimated to be ±1.5%.





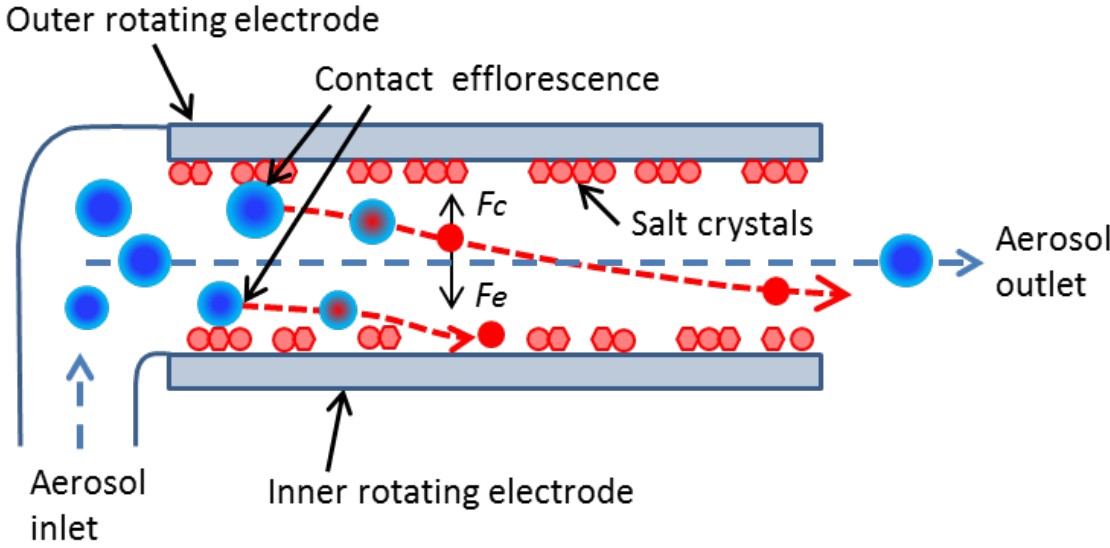

**Figure 5.** Schematic representation of the metastable microdroplets efflorescence when coming into contact with salt crystals deposited on the electrode surface. Explanation and designations are given in the text.



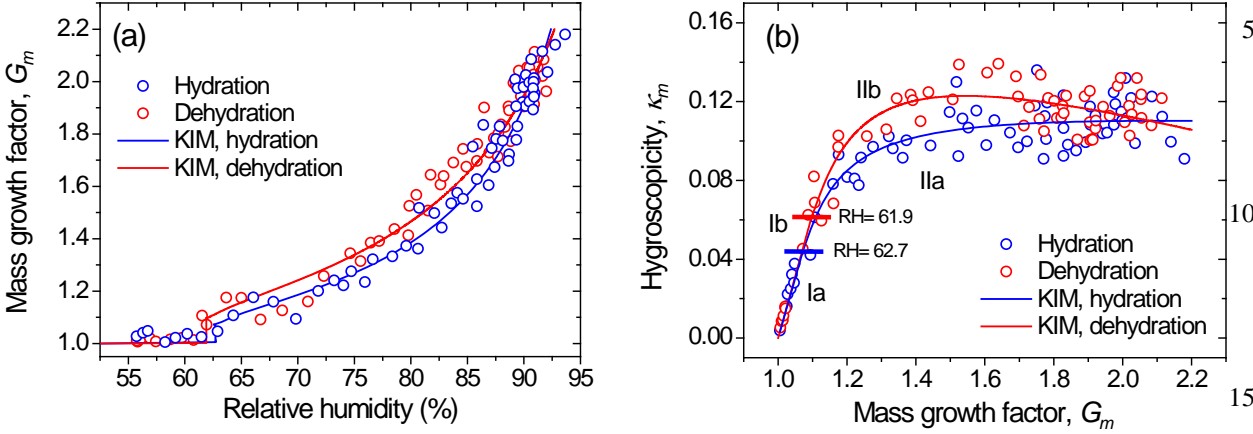

**Figure 6** Hygroscopic properties of ambient atmospheric aerosols with preset $m_d$ =3.5 fg ($D_b$ =160 nm): **(a)** mass growth factor ($G_m$) observed as a function of relative humidity compared to KIM; **(b)** mass-based hygroscopicity parameter ($\kappa_m$) calculated as a function of mass growth factor. The data points are from two repetitive experiments of hydration (blue circles) and dehydration (red circles). The lines are fits of KIM Eqs. (5) and (6). The labels I (Ia,Ib), II (IIa, IIb) indicate different regimes of hygroscopicity (Eqs. (5) and (6)); the borders of the corresponding fit intervals are indicated by blue (hydration) and red (dehydration) bars **(b)**.