# Peer review of "Tandem configuration of differential mobility and centrifugal particle mass analyzers for investigating aerosol hygroscopic properties"

_Atmospheric Measurement Techniques, 2016_

## Referee Comment (RC1) · PH McMurry (Referee) · 13 Oct 2016

The authors propose to use tandem DMA-HCPMA measurements to measure hygroscopic properties of particles. The paper includes a good discussion of the pros and cons of competing methodologies, and clearly explains conceptual advantages of the measurement method they describe. It is an appealing approach.

Nevertheless, I have two major concerns and some minor ones:

Major concerns:

(1) My major concern is the cavalier discussion of measurement technique fundamentals. Figure 2 shows particle number mass distributions with no explanation as to how

they were obtained. In fact, given the relatively broad transfer function of instruments such as the CPMA, obtaining number mass distributions is not straightforward. The literature includes some pertinent information, which is not discussed.

Park et al. (Park et al. 2003) reported on measurements of aerosol mass distributions as a function of mobility diameter from DMA-APM measurements. Their method assumed that particles classified by the DMA had only a single mass (the "modal mass" in the language of the paper under review), which is only approximately true even for chemically homogeneous aerosols. Because the transfer function of a DMA is triangular, mobility-classified particles have a distribution of masses and that distribution affects APM (or CPMA) data. The transfer function of the APM (or CPMA) is even broader than that of the DMA, so at the voltage corresponding to the "modal mass" some particles of every size leaving the DMA penetrate through the APM (or CPMA). Furthermore, ambient aerosols of a given mobility size may include particles of that are chemically and morphologically distinct, which leads to multimodal mass distributions. These subtleties need to be acknowledged in a measurement techniques paper.

More recently, Rawat and coworkers (Rawat et al. 2016) developed an inversion algorithm for obtaining two dimensional number distributions (as a function of mobility diameter and mass) from DMA-APM measurements. Equation 2 of that paper shows the relationship between measured number concentrations downstream of the APM, and operating characteristics of the DMA-APM apparatus (flow rates, voltages, etc., which determine the DMA & APM transfer functions.) Extending their approach to DMA-CPMA data should be possible provided the CPMA transfer function is sufficiently well known. However, Vlasenko and coworkers do not discuss this conceptually important background. I suspect the number mass distributions shown in Figures 2 & 4 were obtained by assuming that the number mass distribution was constant at a given CPMA classifying voltage. Given the narrowness of the sampled aerosol distribution and the breadth of the CPMA transfer function, that is a not a good assumption.

Most previous DMA-APM (or CPMA) work has involved working with raw data: i.e.

measurements of number concentration downstream of the CPMA as a function of CPMA classifying voltage (or equivalently, modal mass), and this is a valid approach. If the authors choose not to use a mathematically justified approach for inverting data to obtain number distributions with respect to mass, I would recommend that they stick to analysis of the raw data. This would involve revising figures 2 & 4 to show only N versus VCPMA, or equivalently, N versus modal mass. The figures 3 & 6 are based on the modal mass (i.e., the value that corresponds to the peak value in the N(VCPMA) measurements), so as far as I can tell the distribution functions are not required for the analyses that were done.

(2) While I this methodology is conceptually appealing, I do not believe the paper delivers on the abstract's promise: "The direct measurements of humidified particle mass allow avoiding complications that occur in the commonly used mobility-diameter-based HTDMA technique due to poorly defined particle morphology and density." It is clear from results of the paper that heating within the CPMA and the broad transfer function of the CPMA lead to complications that are at least as great as those that occur with the HTDMA. The abstract fails to provide a straightforward assessment of the proposed measurement technique's weaknesses. The abstract needs to be forthright about identifying those weaknesses.

Minor Concerns:

(1) Based on results presented in the paper, I think a strong case can be made that the HTDMA method is in principle better for measuring deliquescence and efflorescence thresholds. It is easier to operate a HTDMA under isothermal conditions.

(2) On p. 5 it is stated ". . . for AS and NaCl particles with initial mobility diameter Db=60 nm and dry masses of 0.18±0.01 fg and 0.21±0.01 fg." I assume "±0.01" corresponds to the estimated uncertainty in the modal mass. While the modal mass may be known with high certainty, the relatively broad transfer function of the CPMA ensures that the range of masses exiting the CPMA greatly exceed this value. The authors need to

explain why the modal mass is the proper variable even though sampled mass distributions might have been multimodal (see point 3 below).

(3) The abscissas of Figures 2 & 4 are labeled "Particle Mass". I recommend they be relabelled "Modal Mass". The reader needs to understand that, in fact, particles covering a broad range of masses were present at each CPMA voltage. The importance of this is emphasized by the discussion on p. 6 "..the output aerosol in dehydration mode is a mixture of droplets and dry particles.." If the CPMA transfer function were sufficiently narrow, it would have been possible to distinguish between droplets and dry particles. It is also possible (but not guaranteed) that his could have been achieved if an inversion method similar to that discussed by Rawat et al. had been used to retrieve the true mass distribution. In any event, this phrase supports my argument that these plots do not show mass distributions and need to be replotted.

(4) I am confused by Figure 3. For AS, the blue "+" is labelled "non-prompt efflorescence" while for NaCl, the blue "+" is labelled "Non -prompt deliquescence". The text on p. 5 states "For both AS and NaCl particles, intermediate growth factors between dry and deliquesced particles were observed (Fig. 3 - blue crosses)." The text contradicts the figure label.

(5) I am not convinced that contact efflorescence explains the results and that this might be an approach for studying contact efflorescence (see Figure 5 and discussion on p. 6). Wouldn't it be possible to test this idea by carrying out measurements extending from high voltages, where all particles reach the inner rotating electrode, to low voltages, where all particles reach the outer rotating electrode? The proportion of particles undergoing contact efflorescence should be higher at the low or high voltages, right? Is there any evidence for this?

(6) Does the extent of non-prompt efflorescence and deliquescence change if measurements are carried out when the CPMA is first turned on (i.e., before frictional heating has had time to warm it up)?

In summary, the proposed measurement methodology offers clear conceptual benefits over other methods such as the HTDMA for studying particle phase transitions and hygroscopicity. However, the measurements that are reported reveal limitations on measurement accuracy that may difficult to overcome. Furthermore, I question the validity of Figures 2 & 4. Because the method, in principle, adds to what can be learned from other techniques, I feel it would merit publication after the authors respond to the points raised above.

Park, K., D. B. Kittelson and P. H. McMurry (2003). "A closure study of aerosol mass concentration measurements: comparison of values obtained with filters and by direct measurements of mass distributions." Atmospheric Environment 37(9-10): 1223-1230. Rawat, V. K., D. Buckley, S. Kimoto, M.-H. Lee, N. Fukushima and C. J. Hogan Jr. (2016). "Two-dimensional size-mass distribution function inversion from differential mobility analyzer-aerosol particle mass analyzer (DMA-APM) measurements." Jounrnal of Aerosol Sci. 92: 70-82.

---

## Referee Comment (RC3) · Anonymous Referee #3 · 20 Nov 2016

The authors outline using a differential mobility analyzer and centrifugal particle mass analyzer to measure the mass-based hygroscopicity of nanoparticles. The overall structure of the paper is appropriate; summarizing previous hygroscopic methodologies, validating the proposed system against previous literature values and using the system to characterize an atmospheric aerosol.

Major Comments:

1. Since the paper is describing a relatively new methodology, greater detail is required to explain the advantages and drawbacks of the proposed system, including:

a. A discussion on the effects of multiply-charged particles, system resolution and

the effect of the experimental design parameters. For example was the intention of introducing the 1 lpm of humidified air prior to the CPMA inlet to decrease the residence time of the aerosol in the CPMA and thus reduce heating effects? The reduction in CPMA classifier resolution due to this increased flowrate should be mentioned.

b. The resolution the CPMA was operated at and how it compares to the width of the unclassified particle size distributions measured. It would be beneficial to explain that operating at higher resolutions, requires higher rotational speeds and thus additional heating effects.

c. A discussion of possible data inversion techniques that could be applied to the mass-to-charge distribution measured by the CPMA and the method that was chosen for this work. For example, asymmetric normal distribution or a lognormal distribution (Tajima et al. 2011; Johnson et al. 2013) or convolution of the DMA and CPMA transfer functions (Emery 2005; Barone et al. 2011).

d. A discussion on the system uncertainty and its propagation into the final results, incorporating DMA and CPMA setpoint uncertainty.

2. The discussion on contact efflorescence on Page 6 assumes that even though the particle contacts a charged surface indirectly through the surface crystals the particle's charge does not transfer and that the CPMA centrifugal force is sufficient to separate the particle from the surface crystals after efflorescence transition. Was there evidence of crystal formation on the CPMA inner electrode during classifier cleaning? Furthermore, if the inner electrode surface crystals grew large enough to alter the CPMA classifier's gap, arcing between the cylinders (especially at higher RHs) would likely become an issue first.

Minor Corrections:

1. Given the CPMA measures a mass-to-charge distribution and the variety of data inversion techniques, each mention of particle mass should be further clarified, such

as changing "mass" on Page 1 Line 16 to "modal mass".

2. On Page 2 Line 16, it should be reflected that the DMA-HCPMA configuration has been utilized by others to measure the mass-based hygroscopicity of nanoparticles, such as Johnson et al. 2015.

3. Referring to Page 14 Line 20, did the temperature of the CPMA stabilize after extended periods of continuous operation? If so, at what temperature? This information would be valuable for others applying this method in the future.

4. On Page 3 Line 23, it would be valuable to mention that similar to the DMA, the upper RH limit of the CPMA is limited by voltage arcing between the classifier cylinders.

References

Barone, T. L., Lall, A. A., Storey, J. M. E., Mulholland, G. W., Prikhodko, V.Y., Frankland, J. H., et al. (2011). Size-Resolved Density Measurements of Particle Emissions from an Advanced Combustion Diesel Engine: Effect of Aggregate Morphology. Energy & Fuels, 25(5):1978–1988.

Emery, M. S. (2005). Theoretical Analysis of Data from DMA–APM System. Master's thesis, Particle Technology Laboratory., Department of Mechanical Engineering, University of Minnesota, USA

Johnson, T. J., Symonds, J. P., & Olfert, J. S. (2013). Mass-mobility measurements using a centrifugal particle mass analyzer and differential mobility spectrometer. Aerosol Science and Technology, 47(11), 1215-1225.

Johnson, T. J.; Olfert, J. S.; Yurteri, C. U.; Cabot, R.; & McAughey, J. (2015). Hygroscopic effects on the mobility and mass of cigarette smoke particles. Journal of Aerosol Science, 86, 69-78.

Tajima, N., Fukushima, N., Ehara, K., and Sakurai, H. (2011). Mass Range and Optimized Operation of the Aerosol Particle Mass Analyzer. Aerosol Sci. Technol.,

45(2):196–214

---

## Author Comment (AC1) · 13 Jan 2017

We would like to thank Peter McMurry for the constructive criticism and suggestions for improvement that were taken into account upon manuscript revision. Responses to individual comments are given below.

**Major concerns:**
**Comment from Referee (#1).**

*(1) My major concern is the cavalier discussion of measurement technique fundamentals. Figure 2 shows particle number mass distributions with no explanation as to how they were obtained. In fact, given the relatively broad transfer function of instruments such as the CPMA, obtaining number mass distributions is not straightforward. The literature includes some pertinent information, which is not discussed. Park et al. (Park et al. 2003) reported on measurements of aerosol mass distributions as a function of mobility diameter from DMA-APM measurements. Their method assumed that particles classified by the DMA had only a single mass (the "modal mass" in the language of the paper under review), which is only approximately true even for chemically homogeneous aerosols. Because the transfer function of a DMA is triangular, mobility-classified particles have a distribution of masses and that distribution affects APM (or CPMA) data. The transfer function of the APM (or CPMA) is even broader than that of the DMA, so at the voltage corresponding to the "modal mass" some particles of every size leaving the DMA penetrate through the APM (or CPMA). Furthermore, ambient aerosols of a given mobility size may include particles of that are chemically and morphologically distinct, which leads to multimodal mass distributions. These subtleties need to be acknowledged in a measurement techniques paper. More recently, Rawat and coworkers (Rawat et al. 2016) developed an inversion algorithm for obtaining two dimensional number distributions (as a function of mobility diameter and mass) from DMA-APM measurements. Equation 2 of that paper shows the relationship between measured number concentrations downstream of the APM, and operating characteristics of the DMA-APM apparatus (flow rates, voltages, etc., which determine the DMA & APM transfer functions.) Extending their approach to DMA CPMA data should be possible provided the CPMA transfer function is sufficiently well known. However, Vlasenko and coworkers do not discuss this conceptually important background. I suspect the number mass distributions shown in Figures 2 & 4 were obtained by assuming that the number mass distribution was constant at a given CPMA classifying voltage. Given the narrowness of the sampled aerosol distribution and the breadth of the CPMA transfer function, that is a not a good assumption. Most previous DMA-APM (or CPMA) work has involved working with raw data: i.e measurements of number concentration downstream of the CPMA as a function of CPMA classifying voltage (or equivalently, modal mass), and this is a valid approach. If the authors choose not to use a mathematically justified approach for inverting data to obtain number distributions with respect to mass, I would recommend that they stick to analysis of the raw data. This would involve revising figures 2 & 4 to show only N versus VCPMA, or equivalently, N versus modal mass. The figures 3 & 6 are based on the modal mass (i.e., the value that corresponds to the peak value in the N(VCPMA) measurements), so as far as I can tell the distribution functions are not required for the analyses that were done.*

**Response**
We acknowledge that some questions need more detailed explanation and some additions were inserted in the text. But we believe that the CPMA transfer function is narrower than the DMA transfer function (geom. st. dev. 1.03 against 1.05 in size scale) which determines the width of

input particles distribution. Besides we applied inversion procedure to our data as recommended by the all referees.

50 **Change in manuscript**

The following fragments are added to manuscript in response to comments:

"Thus, a CPMA selects particles with a mass ($m^*$), provided that the charge on the particles is the same and known (Olfert et al., 2006)

$$m^* = \frac{zeV}{\omega_c^2 r_c^c \ln(r_2/r_1)} ,$$
(1)

55

where $V$ is the voltage between inner and outer cylinders with radii $r_1$ and $r_2$ , $z$ is the number of elementary charges $e$ on the particles, $r_c$ = $(r_1 + r_2 )/2$ – centre radius, and $\omega$ is angular velocity at $r_c$ . To improve the transfer function of the classifier, the outer electrode rotates slightly faster than the inner one, producing a stable system of forces (Olfert and Collings, 2005).

60 The particle mass analyzer was operated in the step-by-step scanning mode, where rotation speed and applied voltage are varied in a discrete way to scan the desirable particle mass range. The CMPA, in conjunction with the condensation particle counter (CPC) (TSI model 3787), measured the particle mass based spectrum as a function of the applied RH history. At each step in the scanning mode the detector (CPC) registers the total particle concentration $\Delta N$ passed

65 through the CPMA . This concentration mainly depends on the width and the amplitude of the CPMA transfer function which is essentially triangular in case of neutral stability. The mass setpoint defined by (1) correspond to the centre of the transfer function. The width $\Delta m$ of the function at the half-maximum level determines the mass resolution of the CPMA. In scanning mode the resolution parameter of the CPMA, $R=m^*/\Delta m$ is automatically maintained at the

70 preset value. Therefore, the CPMA provides the averaged mass spectral density - $\Delta N/\Delta m$ or in logarithmic scale $\Delta N/\Delta log (m)= \Delta N/log(1+1/R_m)$. The resolution parameter of the CPMA depends on voltage, rotational rate, air flow and indirectly upon desirable mass range. Its selection is a compromise between the contradictory conditions. For example the high resolution requires rapid electrodes rotation and heightened voltage that increased heat producing and risk

75 of discharge inside the CPMA. In the present work we used by default R = 5, that corresponds to geometric standard deviation 1.08 and 1.03 in the mass and size scales respectively".

"Obviously the concept of the described method is quite identical to widely used HTDMA technique. This approach deals only with modal values of relatively narrow distributions, that

80 makes it less sensitive to the effects of such instrumental factors as transport losses, detection efficiency and multiple charging . Following Rawat et al. (2016) and Stolzenberg & McMurry (2008) the registered particles concentration can be linked to mass-based distribution function $dn/dm$ through the equation :

$$\Delta N(m_i) = \sum_{z=1}^{\infty} \int_0^{\infty} \varepsilon(m)\Theta(z, m, m_i)f(z, m)\frac{dn}{dm}dm \qquad ,$$
(3)

85 where $i$ is a number of the step in the CPMA scanning mode, $m_i$ and $\Theta$ are the mass setpoint and the respective transfer function, $f(z,m)$ is the fraction of particles of mass $m$ with $z$ elementary charges, $\varepsilon(m)$ is transport efficiency through system tubing. In most of our experiments the particles distribution was rather narrow with mass geometric standard deviation of about 1.10 which is slightly more than mass geometric standard deviation of the CPMA

90 transfer function. Firstly it means a clear resolution of peaks of multiple-charged particles (Symonds et al., 2011; McMurry et al., 2002). For particles passed through the DMA with mobility diameter setpoint $D_b$=70 nm the registered by the CPMA the double to single charged particles mass ratio is about 1.7 that is considerably larger than the width of the particles

distribution as well as the CPMA transfer function. Secondly the variations in $\varepsilon(m)$ and $f(z,m)$
95   across the width of distribution function are relatively small that means negligible shift in
position of maximums of $\Delta N/\Delta m$ and $dn/dm$ though their amplitude values and widths are
different.

The Twomey-Markowski algorithm (Markowski 1987; Alofs & Balakumar 1982) was applied to
inverse the equation (3) and estimate the mass-based distribution function as described in detail
100   in supplemental information to Rawat et al. (2016). We used provided there equations for
transport and detection efficiency converted in mass scale. For deconvolution we employed the
idealized triangular transfer function recommended by the manufacturer and measured by
Olfert et al. (2006). The results are shown in Fig.2 (dash curves). The deconvoluted functions
are narrower than experimental distributions but the modal mass values of $\Delta N/\Delta m$ and $dn/dm$
105   agree within 2%. This inversion procedure was applied to the CPMA measurements though we
consider it is not critical in this study. Some exceptions are discussed below.

[Figure]

**Figure 2**. HCPMA measured particle number mass distribution of ammonium sulfate at different RH with initial dry
particle modal mass $m_d = 0.18$ fg. The indicated mode is the modal value of the particle mass distribution used for
the mass growth factor ($G_m$) calculation. Symbols and solid lines – experimental averaged mass spectral density
$\Delta N/\Delta \log (m)$. Dashed lines - mass-based distribution function after application of inversion procedure to primary
data.

The precision of the CPMA particle mass measurements mainly depends on the uncertainties of
voltage , rotation speed, air flow rate and profile between electrodes. The voltage and speed are
110   software controlled inside the CPMA within 0.02% and registered in data output files.
Calculated from this data (using Eq.(1)) the mass setpoint uncertainty was less than 0.1%. The
air flow rate seems the most unstable factor which fluctuated within 2-3 %. The flow rate affects
the CPMA resolution and not the mass setpoint, so its contribution to the mass uncertainty is
difficult to account. Practically the mass uncertainty determined as standard deviation of
115   repeated measurements that took into account the DMA setpoint uncertainty as well. There were
a lot of dry aerosol measurements distributed throughout the experimental period and for dry
aerosol the mass uncertainty was 5% that agree with the results of other researches (McMurry et
al., 2002; Joynson et al., 2015). The number of repeated measurements at a certain RH is not so
large and though the measured mass usually were scattered within 5% we assumed the mass
120   uncertainty in humid conditions equal to the transfer function width (8%). According to Eq.(2b)
this uncertainties translates into a 10% uncertainty in $G_m$"

**Comment from Referee (#2)**

*(2) While I this methodology is conceptually appealing, I do not believe the paper delivers on the abstract's promise: "The direct measurements of humidified particle mass allow avoiding complications that occur in the commonly used mobility-diameter-based HTDMA technique due to poorly defined particle morphology and density." It is clear from results of the paper that heating within the CPMA and the broad transfer function of the CPMA lead to complications that are at least as great as those that occur with the HTDMA. The abstract fails to provide a straightforward assessment of the proposed measurement technique's weaknesses. The abstract needs to be forthright about identifying those weaknesses.*

**Response**

We believe that heating is not a crucial complication and its effect can be compensate by different means. Perhaps our way is not optimal, but it provides satisfactory results. The breadth of the CPMA transfer function depends on operational parameters and can vary within certain range. In some operation modes it is narrower than the DMA transfer function. So we believe it is not an inherent weakness of the technique to note it in the abstract. We added some phrases relating to that in main text.

**Change in manuscript**

"The width $\Delta m$ of the function at the half-maximum level determines the mass resolution of the CPMA. In scanning mode the resolution parameter of the CPMA, $R=m^*/\Delta m$ is automatically maintained at the preset value Therefore, the CPMA provides the averaged mass spectral density - $\Delta N/\Delta m$ or in logarithmic scale $\Delta N/\Delta log\ (m)= \Delta N/log(1+1/R_m)$. The resolution parameter of the CPMA depends on voltage, rotational rate, air flow and indirectly upon desirable mass range. Its selection is a compromise between the contradictory conditions. For example the high resolution requires rapid electrodes rotation and heightened voltage that increased heat producing and risk of discharge inside the CPMA. In the present work we used the default R = 5, that corresponds to geometric standard deviation 1.08 and 1.03 in the mass and size domains respectively."

**Minor Concerns:**

**Comment from Referee (#3)**

*(1) Based on results presented in the paper, I think a strong case can be made that the HTDMA method is in principle better for measuring deliquescence and efflorescence thresholds. It is easier to operate a HTDMA under isothermal conditions.*

**Response**

We agree that currently DRH and ERH measuring with the CPMA is worse than those obtained by HTDMA method due to the relatively large RH uncertainty. We are working on this issue.

**Comment from Referee (#4)**

*(2) On p. 5 it is stated"... for AS and NaCl particles with initial mobility diameter Db=60 nm and dry masses of 0.18±0.01fg and 0.21±0.01fg."I assume"±0.01"corresponds to the estimated uncertainty in the modal mass. While the modal mass may be known with high certainty, the relatively broad transfer function of the CPMA ensures that the range of masses exiting the CPMA greatly exceed this value. The authors need to explain why the modal mass is the proper variable even though sampled mass distributions might have been multimodal (see point 3 below).*

**Response**

We have developed the technique assuming narrow unimodal input distribution. In this case the modal mass is a reasonable parameter to describe the hygroscopic growth and it can be determined within 5%. The cases where particles distributions proved to be broad or even

170    multimodal need more careful consideration. Of course the positions of peaks are determined with more uncertainty. We tried to explain that in those cases where the results are not very reliable.

**Comment from Referee (#5)**

175    *(3) The abscissas of Figures 2 & 4 are labeled "Particle Mass". I recommend they be relabelled "Modal Mass". The reader needs to understand that, in fact, particles covering a broad range of masses were present at each CPMA voltage. The importance of this is emphasized by the discussion on p. 6 "..the output aerosol in dehydration mode is a mixture of droplets and dry particles.." If the CPMA transfer function were sufficiently narrow, it would have been possible*

180    *to distinguish between droplets and dry particles. It is also possible (but not guaranteed) that his could have been achieved if an inversion method similar to that discussed by Rawat et al. had been used to retrieve the true mass distribution. In any event, this phrase supports my argument that these plots do not show mass distributions and need to be replotted.*

**Response**

185    Following this recommendation we applied inversion procedure to our data and corrected the terminology. Changes in the text have described above.
The abscissas of Fig.2 and Fig.4 were relabelled as "CPMA mass setpoint (fg)" .

**Comment from Referee (#6)**

*(4) I am confused by Figure 3. For AS, the blue "+" is labelled "non-prompt efflorescence "while*

190    *for NaCl ,the blue"+"is labeled "Non-prompt deliquescence". The text on p. 5 states "For both AS and NaCl particles, intermediate growth factors between dry and deliquesced particles were observed (Fig. 3 - blue crosses)." The text contradicts the figure label.*

**Response**

That is a sad mistake. Fig. label is corrected.

195

**Comment from Referee (#7)**

*(5) I am not convinced that contact efflorescence explains the results and that this might be an approach for studying contact efflorescence (see Figure 5 and discussion on p. 6). Wouldn't it be possible to test this idea by carrying out measurements extending from high voltages, where all*

200    *particles reach the inner rotating electrode, to low voltages, where all particles reach the outer rotating electrode? The proportion of particles undergoing contact efflorescence should be higher at the low or high voltages, right? Is there any evidence for this?*

**Response**

Contact efflorescence is considered as a possible reason for the observed bimodal

205    distributions. This is only a hypothesis. To confirm or disprove this assumption one needs to fulfill a special study that beyond the issue of the paper. A proposal to vary voltage seems promising but difficult to implement because voltage related to rotation speed, mass resolution and so on. We believe that trajectories of particles need to be simulated inside the CPMA taking into account the possibility of contact efflorescence.

210    **Change in manuscript**

The following clarifying sentence has been added:
"It should be noted that contact efflorescence inside CPMA was suggested as the most plausible explanation for the observed early ERH. Additional experimental and modelling studies are needed to test this hypothesis".

215    To avoid misunderstandings, in conclusion the following text has been removed:
"We suggest that under controlled composition of particles on the electrode surface, the HCPMA system could be additionally used to study isochemical and heterochemical contact efflorescence".

**Comment from Referee (#8)**

*(6) Does the extent of non-prompt efflorescence and deliquescence change if measurements are carried out when the CPMA is first turned on (i.e., before frictional heating has had time to warm it up)?*

**Response**

There were a few measurements of efflorescence and deliquescence at the beginning of operation. We rewieved our data from this point of view but failed to reveal any dependence on the temperature inside the CPMA.

**Comment from Referee (#9)**

In summary, the proposed measurement methodology offers clear conceptual benefits over other methods such as the HTDMA for studying particle phase transitions and hygroscopicity. However, the measurements that are reported reveal limitations on measurement accuracy that may difficult to overcome. Furthermore, I question the validity of Figures 2&4. Because the method, in principle, adds to what can be learned from other techniques, I feel it would merit publication after the authors respond to the points raised above.

Park, K., D. B. Kittelson and P. H. McMurry (2003). "A closure study of aerosol mass concentration measurements: comparison of values obtained with filters and by direct measurements of mass distributions." Atmospheric Environment 37(9-10): 1223-1230. Rawat, V. K., D. Buckley, S. Kimoto, M.-H. Lee, N. Fukushima and C. J. Hogan Jr. (2016). "Two-dimensionalsize-massdistributionfunctioninversionfromdifferentialmobilityanalyzer-aerosolparticlemassanalyzer(DMA-APM)measurements."Jounrnalof Aerosol Sci. 92: 70-82. Interactive comment on Atmos. Meas. Tech. Discuss., doi:10.5194/amt-2016-249, 2016.

**Response**

All concerns have been accounted for and additional literature was examined and included in the reference list.

---

## Author Comment (AC2) · 13 Jan 2017

We would like to thank Referee #2 for the constructive criticism and suggestions for improvement that were taken into account upon manuscript revision. Responses to individual comments are given below.

**Major comments from Referee (#1)**

*A more thorough study into the uncertainties of measurement would be ideal, perhaps even as an appendix or SI. Though not common (arguably it should be), error propagation for measurement data would be very helpful in drawing conclusions as to the benefits or drawbacks of measurement techniques. How accurate is a standard HTDMA, for example? Few papers involving the use of HTDMA quantify the errors, yet draw conclusions regarding cloud formation and aerosol hygroscopicity, which in turn propagate through to $\kappa$ values and into models. This is not good practice, and it would be useful to at least include some kind of error propagation (e.g. flows, voltages, etc etc) through to final values. The authors state that the difference between experiment and theoretical growth factor values does not exceed 10%. If the uncertainties are above 10% for the system (likely), then this is indeed good agreement. The authors make no mention of the ZSR assumption of volume additivity. This seems like a near-ideal experimental setup to more accurately probe this conventional assumption. Perhaps ideally, dual-CPMA or CPMA-HDMA would be employed as the first DMA is still dependent on shape factor and morphology whereas particle mass is always particle mass. Comment?*

**Response**

It is a fair comment and we tried to pay more attention to errors analysis. But we still believe that the HTDMA errors are beyond the scope of this article. We are planning to combine HTDMA and HCPMA, so the required analysis may be the topic of the future paper.

**Change in manuscript**

The following text is added.

"The precision of the CPMA particle mass measurements mainly depends on the uncertainties of voltage, rotation speed, air flow rate and profile between electrodes. The voltage and speed are software controlled inside the CPMA within 0.02% and registered in data output files. Calculated from this data (using Eq.(1)) the mass setpoint uncertainty was less than 0.1%. The air flow rate seems the most unstable factor which fluctuated within 2-3 %. The flow rate affects the CPMA resolution and not the mass setpoint, so its contribution to the mass uncertainty is difficult to account. Practically the mass uncertainty determined as standard deviation of repeated measurements that took into account the DMA setpoint uncertainty as well. There were a lot of dry aerosol measurements distributed throughout the experimental period and for dry aerosol the mass uncertainty was 5% that agree with the results of other researches (McMurry et al., 2002; Joynson et al., 2015). The number of repeated measurements at a certain RH is not so large and though the measured mass usually were scattered within 5% we assumed the mass uncertainty in humid conditions equal to the transfer function width (8%). According to Eq.(2b) this uncertainties translates into a 10% uncertainty in $G_m$"

**Minor comments:**

**Comment from Referee (#2)**

*Final sentence of the abstract is muddled. It would read better as: "Direct measurements of particle mass avoid the typical complications associated with the commonly used mobility-diameter based HTDMA technique (mainly due to poorly defined or unknown morphology and density)."*

**Response**

We agree with this observation and make the proposed revisions

**Change in manuscript**

55    The last sentence of the abstract is replaced by "Direct measurements of particle mass avoid the typical complications associated with the commonly used mobility-diameter based HTDMA technique (mainly due to poorly defined or unknown morphology and density)."

**Comment from Referee (#3)**
60    *Page 2 Line 6: arguably the aerosol is quasi-monodisperse and not truly monodisperse, due to multiple charging.*
Response
We agree that truly monodisperse aerosol is a kind of idealization and never found in nature or laboratory. But we believe that words "monodisperse aerosol" are often used approximately for
65    brevity.
Nevertheless some clarification will be added to the text.
**Change in manuscript**
"The selected particles are not truly monodisperse due to multiple charging effects, non-ideality of DMA transfer function etc. However the width of output size distribution is small enough to
70    consider the aerosol as "quasi-monodisperse.""

**Comment from Referee (#4)**
*Page 2 Line 32: This would be the ideal location to introduce the CPMA in more detail. E.g. "Mass classifiers started with the APM and improved with the CPMA" – the CPMA has a higher*
75    *penetration efficiency for any given mass than the APM*
Response
Perhaps the referee is absolutely right, but we have no practical APM experience and so we are not competent enough to compare the APM and CPMA in detail. We believe that the almost twenty-year history of the CPMA deserves a detailed and thorough review written by researches
80    who have extensive experience with the technique.

**Comment from Referee (#5)**
*Page 3 Line 1: Please check Kondo et al. 2006 for more references; and note that McMurray et al. 2002 conclude the DMA-APM to have around 5% uncertainty (this paper is already*
85    *referenced)*
*Response*
We thank the referee for useful reference.
**Change in manuscript**
"Kondo et al. (2006) mounted the heater upstream the DMA and used the DMA+APM technique
90    to measure a relationship between mass and size of non-volatile particles in ambient air."

**Comment from Referee (#6)**
*Page 3 Line 4: Novel application*
Response
95    Done

**Comment from Referee (#7)**
*Page 3 Line 19: quasi-monodisperse rather than near-monodisperse*
Response
100    The word "near-monodisperse" is used in literature (Geller et al., 2006 in reference list of discussion paper), but we assume that the "quasi-monodisperse" is more appropriate.

**Comment from Referee (#8)**
*Page 4 Line 5: technically, the CPMA classifies particles by their mass: charge ratio, not merely*
105    *mass*
Response
Changed as suggested.
**Change in manuscript**

"according to their  mass to charge ratio"

110

**Comment  from Referee (#9)**

*Page 4 Line 11: the CPMA outer cylinder spins 3% faster ((33/32)-1)*100=3.125 %)*

R**esponse**

That is a sad mistake. The text was corrected.

115

**Comment  from Referee (#10)**

*Page 4 Line 15: arguably, this is not actually a number-mass distribution. It is a distribution of average masses. No inversion has been performed. This terminology should be thoughtfully considered throughout the manuscript.*

120 **Response**

We agree that terminology needs elaboration. To improve the terminology we introduced  "the averaged mass spectral density",,"the mass setpoint", " the center of the transfer function". Besides we applied inversion procedure to our data.

**Change in manuscript**

125 The following text is added:

"Thus, a CPMA selects particles with a mass ($m^*$), provided that the charge on the particles is the same and known (Olfert et al., 2006)

$$m^* = \frac{zeV}{\omega_c^2 r_c^c \ln(r_2/r_1)},$$ (1)

130 where $V$ is the voltage between inner and outer  cylinders  with  radii  $r_1$ and  $r_2$ ,  $z$ is the number of elementary charges $e$  on the particles, $r_c = (r_1 + r_2 )/2$ – centre  radius, and  $\omega$  is angular velocity  at  $r_c$ . To improve the transfer function of the classifier, the outer electrode rotates slightly faster than the inner one, producing a stable system of forces (Olfert and Collings, 2005). The particle mass analyzer was operated in the step-by-step scanning mode, where rotation speed

135 and applied voltage are varied in a discrete way to scan the desirable particle mass range. The CMPA, in conjunction with the condensation particle counter (CPC) (TSI model 3787), measured the particle  mass based spectrum  as a function of the applied RH history. At each step in the scanning  mode the detector (CPC) registers the total particle concentration  $\Delta N$   passed through the CPMA . This concentration mainly depends on the width and the amplitude of the

140 CPMA transfer function which is essentially triangular in case of neutral stability.  The mass setpoint  defined by (1) correspond to the centre of the transfer function. The width $\Delta m$  of  the function at the half-maximum level determines the mass resolution of the CPMA. In scanning mode  the resolution parameter of the CPMA, $R=m^*/\Delta m$ is automatically maintained  at the preset value.  Therefore, the CPMA provides the averaged mass spectral density  - $\Delta N/\Delta m$ or in

145 logarithmic scale  $\Delta N/\Delta log (m)= \Delta N/log(1+1/R_m)$. The resolution parameter of the CPMA depends on voltage, rotational rate, air flow and indirectly upon desirable mass range. Its selection is a compromise between the contradictory conditions. For example the high resolution requires rapid electrodes rotation and heightened voltage that increased heat producing and risk of discharge inside the CPMA.  In the present work we used by default R = 5, that corresponds to

150 geometric standard deviation  1.08 and 1.03  in the mass and size scales  respectively".

**Comment  from Referee (#11)**

*Page 4 Line 25: "were directly entered" should be "entered"*

155 R**esponse**

The text is revised as recommended.

**Comment  from Referee (#12)**

*Page 5 Line 13: Though I understand it's common, it's sloppy/confusing to write AS for*

160 *ammonium sulphate, as that would make sodium chloride SC. (NH4)2SO4 would be correct.*

R**esponse**

Changed as recommended except Sec.3 where AS used as subscript.

**Comment from Referee (#13)**

*Page 5 Line 26: Not enough emphasis is placed on mass error of 10% translating to size error of 3%. This is where the DMA-HCPMA technique has the potential for significant benefit over some HTDMA measurements, should other issues be addressed.*

Response

We agree that HCPMA has some potential advantages but consider it would better to emphasis after gaining more experience in practical measurements.

**Comment from Referee (#14)**

*Page 6 Line 6: . . . artefact of this technique*

Response

The text is revised as recommended

**Comment from Referee (#15)**

*Page 6 Line 17: It is important to note that, as with the DMA, any change of the particle water uptake. . .*

*Response*

The text is revised as recommended

**Further comments (#16):**

*Though it is really the place of the authors to address referee comments, the first Referee is not correct in stating that the "transfer function of the APM or CPMA is even broader than the DMA", and this should be addressed. The transfer function of an instrument is the relationship between its input and its output. For the CPMA, it relates the mass setpoint to the ratio between the concentration of aerosol downstream and upstream of the CPMA. The resolution of the CPMA is set by the width of the transfer function, and the maximum penetration by its height. For the case of neutral stability the transfer function of the CPMA is essentially triangular, and theoretically extends to 100% transmission at its peak (Figure 1):*
*It is of course possible to pre-classify with a DMA, but the highest resolution of the CPMA in size terms is much higher than any available DMA. However, broadening as with HTDMA instruments is non-trivial and should be discussed. I agree that the inversion developed by Rawat et al. (2016) should at least be discussed in this paper. Regarding point (3) raised by Referee 1; the CPMA transfer function is not broad in mass-space. I agree that a broad range of masses enter the CPMA from the DMA, as the DMA's output will cover a range of masses as it's a mobility classifier. I refer the authors to my earlier comment regarding "average particle mass", though "modal mass" is probably more accurate as stated by Referee 1. For proper mass distributions the inversion by Rawat et al., or similar, should be used.*

Response

We agree with all referees that inversion procedure should be applied to the CPMA data to obtain mass distribution function. But we believe it is not critically needed in our approach because it based on modal mass measurements. Exception – dehydration hysteresis. We decided to employ the inversion algorithm to our data to remove objections.

**Change in manuscript**

The following text is added:

"Obviously the concept of the described method is quit identical to widely used HTDMA technique. This approach deals only with modal values of relatively narrow distributions, that makes it less sensitive to the effects of such instrumental factors as transport losses, detection efficiency and multiple charging. Following Rawat et al. (2016) and Stolzenberg & McMurry (2008) the registered particles concentration can be linked to mass-based distribution function *dn/dm* through the equation :

$$\Delta N(m_i) = \sum_{z=1}^{\infty} \int_0^{\infty} \varepsilon(m)\Theta(z,m,m_i)f(z,m)\frac{dn}{dm}dm \qquad , \qquad (3)$$

where $i$ is a number of the step in the CPMA scanning mode, $m_i$ and $\Theta$ are the mass setpoint and the respective transfer function, $f(z,m)$ is the fraction of particles of mass $m$ with $z$
220 elementary charges, $\varepsilon(m)$ is transport efficiency through system tubing. In most of our experiments the particles distribution was rather narrow with mass geometric standard deviation of about 1.10 which is slightly more than mass geometric standard deviation of the CPMA transfer function. Firstly it means a clear resolution of peaks of multiple-charged particles (Symonds et al., 2011; McMurry et al., 2002). For particles passed through the DMA with
225 mobility diameter setpoint $D_b$=70 nm the registered by the CPMA the double to single charged particles mass ratio is about 1.7 that is considerably larger than the width of the particles distribution as well as the CPMA transfer function. Secondly the variations in $\varepsilon(m)$ and $f(z,m)$ across the width of distribution function are relatively small that means negligible shift in position of maximums of $\Delta N/\Delta m$ and $dn/dm$ though their amplitude values and widths are
230 different.

The Twomey-Markowski algorithm (Markowski 1987; Alofs & Balakumar 1982) was applied to inverse the equation (3) and estimate the mass-based distribution function as described in detail in supplemental information to Rawat et al. (2016). We used provided there equations for transport and detection efficiency converted in mass scale. For deconvolution we employed the
235 idealized triangular transfer function recommended by the manufacturer and measured by Olfert et al. (2006). The results are shown in Fig.2 (dash curves). The deconvoluted functions are narrower than experimental distributions but the modal mass values of $\Delta N/\Delta m$ and $dn/dm$ agree within 2%. This inversion procedure was applied to the CPMA measurements though we consider it is not critical in this study. Some exceptions are discussed below."

240

[Figure]

**Figure 2**. HCPMA measured particle number mass distribution of ammonium sulfate at different RH with initial dry particle modal mass $m_d = 0.18$ fg. The indicated mode is the modal value of the particle mass distribution used for the mass growth factor ($G_m$) calculation. Symbols and solid lines – experimental averaged mass spectral density $\Delta N/\Delta log\,(m)$. Dashed lines - mass-based distribution function after application of inversion procedure to primary data.

---

## Author Comment (AC3) · 13 Jan 2017

5   We would like to thank Referee #3 for the constructive criticism and suggestions for improvement that were taken into account upon manuscript revision. Responses to individual comments are given below.

**Major Comments (#1):**
10  *1. Since the paper is describing a relatively new methodology, greater detail is required to explain the advantages and drawbacks of the proposed system, including:*
*a. A discussion on the effects of multiply-charged particles, system resolution and the effect of the experimental design parameters. For example was the intention of introducing the 1 lpm of humidified air prior to the CPMA inlet to decrease the*
15  *residence time of the aerosol in the CPMA and thus reduce heating effects? The reduction in CPMA classifier resolution due to this increased flowrate should be mentioned.*
*b. The resolution the CPMA was operated at and how it compares to the width of the unclassified particle size distributions measured. It would be beneficial to explain*
20  *that operating at higher resolutions, requires higher rotational speeds and thus additional heating effects.*
*c. A discussion of possible data inversion techniques that could be applied to the mass-to-charge distribution measured by the CPMA and the method that was chosen for this work. For example, asymmetric normal distribution or a lognormal*
25  *distribution (Tajima et al. 2011; Johnson et al. 2013) or convolution of the DMA and CPMA transfer functions (Emery 2005; Barone et al. 2011).*
*d. A discussion on the system uncertainty and its propagation into the final results, incorporating DMA and CPMA setpoint uncertainty.*
R**esponse**
30  We agree with referee and tried to improve the text. The air flow system was designed to provide required RH and residence time that determined the flowrate. The resolution was selected for the given flow rate to decrease heating.
**Change in manuscript**
The following fragments are added to manuscript in response to comments (a-d)
35  "Thus, a CPMA selects particles with a mass ($m^*$), provided that the charge on the particles is the same and known (Olfert et al., 2006)

$$m^* = \frac{zeV}{\omega_c^2 r_c^c \ln(r_2/r_1)},\tag{1}$$

where $V$ is the voltage between inner and outer cylinders with radii $r_1$ and $r_2$, $z$ is the number of elementary charges $e$ on the particles, $r_c = (r_1 + r_2)/2$ – centre radius, and
40  $\omega$ is angular velocity at $r_c$. To improve the transfer function of the classifier, the outer electrode rotates slightly faster than the inner one, producing a stable system of forces (Olfert and Collings, 2005). The particle mass analyzer was operated in the step-by-step scanning mode, where rotation speed and applied voltage are varied in a discrete way to scan the desirable particle mass range. The CMPA, in conjunction with the condensation
45  particle counter (CPC) (TSI model 3787), measured the particle mass based spectrum as a function of the applied RH history. At each step in the scanning mode the detector (CPC) registers the total particle concentration $\Delta N$ passed through the CPMA. This concentration mainly depends on the width and the amplitude of the CPMA transfer function which is essentially triangular in case of neutral stability. The mass setpoint
50  defined by (1) correspond to the centre of the transfer function. The width $\Delta m$ of the function at the half-maximum level determines the mass resolution of the CPMA. In scanning mode the resolution parameter of the CPMA, $R=m^*/\Delta m$ is automatically maintained at the preset value. Therefore, the CPMA provides the averaged mass

spectral density - $\Delta N/\Delta m$ or in logarithmic scale $\Delta N/\Delta log\ (m)= \Delta N/log(1+1/R_m)$. The
resolution parameter of the CPMA depends on voltage, rotational rate, air flow and
indirectly upon desirable mass range. Its selection is a compromise between the
contradictory conditions. For example the high resolution requires rapid electrodes
rotation and heightened voltage that increased heat producing and risk of discharge
inside the CPMA.  In the present work we used by default R = 5, that corresponds to
geometric standard deviation  1.08 and 1.03  in the mass and size scales  respectively".

"Obviously the concept of the described method is quite identical to widely used
HTDMA technique.  This approach deals only with modal values of relatively narrow
distributions, that makes it less sensitive to the effects of such instrumental factors as
transport losses, detection efficiency and multiple charging . Following  Rawat et al.
(2016) and Stolzenberg & McMurry (2008) the measured particles concentration can be
linked to mass-based distribution function $dn/dm$ through the equation :

$$\Delta N(m_i) = \sum_{z=1}^{\infty}\int_0^{\infty}\varepsilon(m)\Theta(z,m,m_i)f(z,m)\frac{dn}{dm}dm \qquad , \qquad (3)$$

where  $i$ is a  number of  the step in the  CPMA scanning  mode,  $m_i$ and  $\Theta$ are the mass
setpoint and the respective transfer function,  $f\,(z,m)$  is the fraction of particles of mass
$m$ with $z$ elementary charges, $\varepsilon(m)$ is transport efficiency through system tubing.  In most
of our experiments the particles distribution was rather narrow with mass geometric
standard deviation of about 1.10 which is slightly more than mass geometric standard
deviation of the CPMA transfer function.  Firstly it means  a clear resolution of  peaks of
multiple-charged particles (Symonds et al., 2011; McMurry et al., 2002).  For particles
passed through the DMA with mobility diameter setpoint  $D_b$=70 nm the registered by
the CPMA  the double to single charged particles mass ratio is about 1.7 that is
considerably large than the width of  the particles distribution  as well as the CPMA
transfer function. Secondly  the variations in  $\varepsilon(m)$ and $f\,(z,m)$  across the  width of
distribution function are relatively small that means negligible shift in position of
maximums of  $\Delta N/\Delta m$ and $dn/dm$  though their amplitude values and widths are
different.
The Twomey-Markowski algorithm (Markowski 1987; Alofs & Balakumar 1982) was
applied to inverse the equation (3) and estimate the mass-based distribution function as
described in detail in supplemental information to Rawat et al. (2016). We used provided
there equations  for transport and detection efficiency converted in the mass  scale. For
deconvolution we employed the idealized triangular transfer function recommended  by
the manufacturer   and  measured by Olfert et al. (2006). The results are shown in Fig.2
(dash curves).  The  deconvoluted functions are narrower than experimental distributions
but  the modal mass values of  $\Delta N/\Delta m$ and $dn/dm$  agree within 2%.  This inversion
procedure was applied to the CPMA measurements though we consider  it is not critical
in this study. Some exceptions are discussed below.
The precision  of the CPMA particle mass measurements  mainly depends on the
uncertainties of  voltage , rotation speed, air flow rate and profile between electrodes.
The voltage and speed are software controlled inside the CPMA within 0.02%  and
registered in data output files. Calculated from this data  (using Eq.(1))  the mass
setpoint uncertainty was less than 0.1%. The air flow rate seems the most unstable factor
which fluctuated within 2-3 %.  The flow rate affects the CPMA resolution  and not the
mass setpoint, so its contribution to the mass uncertainty is difficult to account.
Practically the  mass uncertainty determined  as standard deviation of  repeated
measurements that took into account the DMA setpoint uncertainty as well. There were a
lot of dry aerosol measurements distributed throughout  the experimental period and for
dry aerosol the mass uncertainty was 5% that agree with the results of  other researches
(McMurry et al., 2002; Joynson et al., 2015). The number of repeated measurements at a

certain RH is not so large and though the measured mass usually were scattered within 5% we assumed the mass uncertainty in humid conditions equal to the transfer function width (8%). According to Eq.(2b) this uncertainties translates into a 10% uncertainty in $G_m$ "

110

[Figure]

**Figure 2**. HCPMA measured particle number mass distribution of ammonium sulfate at different RH with initial dry particle modal mass $m_d$ = 0.18 fg. The indicated mode is the modal value of the particle mass distribution used for the mass growth factor ($G_m$) calculation. Symbols and solid lines – experimental averaged mass spectral density $\Delta N/\Delta \log (m)$. Dashed lines - mass-based distribution function after application of inversion procedure to primary data.

**Comment from Referee (#2)**

*2. The discussion on contact efflorescence on Page 6 assumes that even though the*
115 *particle contacts a charged surface indirectly through the surface crystals the*
*particle's charge does not transfer and that the CPMA centrifugal force is sufficient*
*to separate the particle from the surface crystals after efflorescence transition. Was*
*there evidence of crystal formation on the CPMA inner electrode during classifier*
*cleaning? Furthermore, if the inner electrode surface crystals grew large enough*
120 *to alter the CPMA classifier's gap, arcing between the cylinders (especially at*
*higher RHs) would likely become an issue first.*

**Response**

Contact efflorescence is considered as possible reason for the observed bimodal distributions. This is only a hypothesis without any reliable evidence. We agree that
125 charge transfer may happen when particle contacts the electrode or the previously precipitated particle, but the final result of it is uncertain. When cleaning we saw a slight white sediment on the both cylinders (like DMA) but it does not prove anything. The cleaning procedure was regular enough to prevent arcing.

**Change in manuscript**

130 The following clarifying sentence has been added:
"It should be noted that contact efflorescence inside CPMA was suggested as the most plausible explanation for the observed early ERH. Additional experimental and modelling studies are needed to test this hypothesis".
To avoid misunderstandings, in conclusion the following text has been removed:
135 "We suggest that under controlled composition of particles on the electrode surface, the HCPMA system could be additionally used to study isochemical and heterochemical contact efflorescence".

**Minor Corrections (#3):**

140 *1. Given the CPMA measures a mass-to-charge distribution and the variety of data inversion techniques, each mention of particle mass should be further clarified, such*

*as changing "mass" on Page 1 Line 16 to "modal mass".*

**R**esponse

To improve the terminology we introduced where appropriate in text the following terms "the averaged mass spectral density","the mass setpoint", " the center of the transfer function" ," modal mass". Besides we applied inversion procedure to our data.

**Comment from Referee (#4)**

*2. On Page 2 Line 16, it should be reflected that the DMA-HCPMA configuration has been utilized by others to measure the mass-based hygroscopicity of nanoparticles, such as Johnson et al. 2015.*

**R**esponse

Johnson et al. (2015) paid main attention to particles effective density changing due to water uptake. This paper is referenced on page 3, line 3. We believe it would be better to add some words here.

**Change in manuscript**

In the last work the combination of DMA and CDMA was used to measure mass growth factor of tobacco smoke particles as function of increasing RH. This technique is more similar to ours.

**Comment from Referee (#5)**

*3. Referring to Page 14 Line 20, did the temperature of the CPMA stabilize after extended periods of continuous operation? If so, at what temperature? This information would be valuable for others applying this method in the future.*

**R**esponse

After a prolonged continuous operation (3-4 hours) the temperature of the CPMA usually stabilizes at 4-6 degrees above room level depending on average rotation speed. The stable CPMA temperature means the constant temperature of water bath.

**Change in manuscript**

Usually the CPMA temperature stabilized at 4-6 degree above room level after 3-4 hours of continuous operation depending on average rotation speed.

**Comment from Referee (#6)**

*4. On Page 3 Line 23, it would be valuable to mention that similar to the DMA, the upper*

*RH limit of the CPMA is limited by voltage arcing between the classifier cylinders.*

**R**esponse

The voltage between cylinders depends on desirable mass/size range and resolution. The higher voltage is required to classify more massive particles with the same resolution. In most of our experiments the voltage did not exceed 200 V that corresponds electric field 2kV/cm. This field is at least twice less than in DMA, so there was no arcing even at 98% RH. It may be potential advantage of the method. But the high RH is difficult to control and the intensive water condensing in cooler tubes after the CPMA prevents flow stability and proper operation of the CPC. That was the reason of the upper RH limit in our work.